# Effects of In-Hospital Exercise on Frailty in Patients with Hepatocellular Carcinoma

**DOI:** 10.3390/cancers13020194

**Published:** 2021-01-07

**Authors:** Jin Tsuchihashi, Shunji Koya, Keisuke Hirota, Noboru Koga, Hayato Narao, Manabu Tomita, Takumi Kawaguchi, Ryuki Hashida, Dan Nakano, Tsubasa Tsutsumi, Sachiyo Yoshio, Hiroo Matsuse, Taku Sanada, Kazuo Notsumata, Takuji Torimura

**Affiliations:** 1Division of Rehabilitation, Fukui-ken Saiseikai Hospital, Fukui 918-8503, Japan; tsuchihashi.jin8044@fukui.saiseikai.or.jp; 2Division of Rehabilitation, Kurume University Hospital, Kurume 830-0011, Japan; kouya_shunji@kurume-u.ac.jp (S.K.); hirota_keisuke@kurume-u.ac.jp (K.H.); hashida_ryuuki@med.kurume-u.ac.jp (R.H.); matsuse_hiroh@kurume-u.ac.jp (H.M.); 3Department of Rehabilitation, Chikugo City Hospital, Chikugo 833-0041, Japan; koga.nationalhospital@gmail.com; 4Department of Rehabilitation, Yame General Hospital, Yame 834-0034, Japan; hayatonarao198071@gmail.com; 5Department of Rehabilitation, Saga Central Hospital, Saga 849-8522, Japan; sp4u7m99@ray.ocn.ne.jp; 6Division of Gastroenterology, Department of Medicine, Kurume University School of Medicine, Kurume 830-0011, Japan; nakano_dan@med.kurume-u.ac.jp (D.N.); tsutsumi_tsubasa@med.kurume-u.ac.jp (T.T.); tori@med.kurume-u.ac.jp (T.T.); 7Department of Orthopedics, Kurume University School of Medicine, Kurume 830-0011, Japan; 8Department of Liver Disease, Research Center for Hepatitis and Immunology, National Center for Global Health and Medicine, Kohnodai 272-8516, Japan; sachiyo@hospk.ncgm.go.jp; 9Department of General Internal Medicine, Fukui-ken Saiseikai Hospital, Fukui 918-8503, Japan; sanada.taku3054@fukui.saiseikai.or.jp (T.S.); notsumata.kazuo6017@fukui.saiseikai.or.jp (K.N.)

**Keywords:** frailness, physical dysfunction, rehabilitation, hepatoma

## Abstract

**Simple Summary:**

Frailty including physical inactivity is associated with the survival of patients with hepatocellular carcinoma (HCC). We aimed to investigate the effects of in-hospital exercise on frailty in patients with HCC. This was a multi-center observational study. Patients with HCC were classified into exercise (*n* = 114) and non-exercise (*n* = 67) groups. The exercise group was treated with a mixture of aerobic and resistance exercises (20–40 min/day, median four days). Frailty was assessed using the liver frailty index (LFI). In multivariate analysis, exercise (odds ratio (OR) 2.38, 95% confidence interval (CI) 1.240–4.570, *p* = 0.0091) as an independent factor for the improvement of LFI. In the decision-tree analysis, exercise was identified as an initial classifier associated with the improvement of LFI. We demonstrated that in-hospital exercise improved frailty in patients with HCC, which was confirmed by propensity score matching analysis. Thus, in-hospital exercise may be beneficial for improving physical function in patients with HCC.

**Abstract:**

Frailty including physical inactivity is associated with the survival of patients with hepatocellular carcinoma (HCC). We aimed to investigate the effects of in-hospital exercise on frailty in patients with HCC. This was a multi-center observational study. Patients with HCC were classified into exercise (*n* = 114) and non-exercise (*n* = 67) groups. The exercise group was treated with a mixture of aerobic and resistance exercises (20–40 min/day, median four days). Frailty was assessed using the liver frailty index (LFI). Factors for changes in LFI were examined by multivariate and decision-tree analyses. The factors were also examined after propensity score matching. During hospitalization, LFI was significantly improved in the exercise group compared to the non-exercise group (ΔLFI −0.17 vs. −0.02, *p* = 0.0119). In multivariate analysis, exercise (odds ratio (OR) 2.38, 95% confidence interval (CI) 1.240–4.570, *p* = 0.0091) and females (OR 2.09; 95%CI, 1.062–4.109; *p* = 0.0328) were identified as independent factors for the improvement of LFI. In the decision-tree analysis, exercise was identified as an initial classifier associated with the improvement of LFI. Similar findings were also seen in the propensity score matching analyses. We demonstrated that in-hospital exercise improved frailty in patients with HCC. Thus, in-hospital exercise may be beneficial for improving physical function in patients with HCC.

## 1. Introduction

Frailty is a geriatric syndrome of physiological decline that includes physical inactivity [1]. Frailty increases the risk of other geriatric syndromes and adverse health outcomes in older populations [2]. The incidence of frailty is especially high in elder patients with cancer. Over half of elder patients with cancer have frailty or pre-frailty because of both cancer itself and the therapies offered [3]. Frailty has been shown to predict the extent of muscle atrophy [4,5] as well as poorer mental and physical conditions in patients with chronic liver disease [6]. Frailty is also associated with the length of hospital stay in patients with hepatocellular carcinoma (HCC) who underwent orthotopic liver transplantation [7]. In addition, frailty is associated with poor prognosis in patients with HCC [8].

The liver frailty index (LFI) has recently been created to objectively quantify frailty in patients with chronic liver disease [9]. The LFI is calculated using three performance-based physical tests consisting of grip strength, chair stands, and balance [9]. According to the LFI score, patients are classified as robust, pre-frail, or frail [9]. We previously reported that pre-frail/frail determined by LFI was independent of the risk of muscle atrophy in patients with HCC [4]. The LFI predicts muscle atrophy with high sensitivity, even in patients with normal grip strength [4]. In addition, frailty assessed by LFI is reported to be independently associated with liver transplantation waitlist mortality [10]. Thus, LFI scores can be used to objectively quantify the risk of adverse outcomes, including mortality, in patients with chronic liver disease.

Muscle mass is known to significantly decrease during hospitalization for HCC treatment [11]. Exercise is a fundamental treatment for frailty and has been reported to improve physical frailty such as functional/aerobic capacity and sarcopenia in patients with chronic liver disease [12]. In Japan, in-hospital exercise for patients with cancer is an approved health care service by the Japanese Ministry of Health, Labor, and Welfare [13]. Recently, we developed an in-hospital exercise program for patients with HCC, which consists of stretching, strength training, balance practice, and aerobic training [14]. We previously described that in-hospital exercise did not worsen liver function [14] and increased muscle mass in patients with HCC who underwent transcatheter arterial chemoembolization (TACE) [15]. However, the impact of in-hospital exercise on frailty in patients with HCC remains unclear.

This study aimed to investigate the effects of in-hospital exercise on frailty assessed by LFI in patients with HCC.

## 2. Results

### 2.1. Differences in Patient Characteristics between the Exercise and Non-Exercise Groups

Patient characteristics are summarized in Table 1. Patients in the exercise group were significantly older than those in the non-exercise group. There was no significant difference in the female-to-male ratio or BMI between the two groups. No significant difference was seen in the etiology of liver disease, treatment of HCC, or tumor nodes metastasis (TNM) stage between the two groups. There was no significant difference in the albumin-bilirubin (ALBI) grade or the prevalence of branched-chain amino acids (BCAA) supplementation between the two groups. No significant difference was seen in biochemical examinations, including hemoglobin A1c (HbA1c) value and estimated glomerular filtration rate (eGFR), between the exercise and the non-exercise groups (Table 1).

In the exercise group, the LFI score was significantly higher than in the non-exercise group at baseline; however, there was no significant difference in the prevalence of pre-frail/frail between the two groups (Table 1). There was no significant difference in the presence of sarcopenia or visceral fat area (VFA) between the two groups. The hospitalization period was significantly shorter in the exercise group than in the non-exercise group. In the exercise group, the median exercise time was four days and the median exercise implementation rate was 100% (Table 1).

### 2.2. Differences in LFI Changes during Hospitalization between the Exercise and Non-Exercise Groups

Changes in LFI were evaluated based on the differences between the admission and discharge days. The ΔLFI was significantly lower in the exercise group than in the non-exercise group (Figure 1).

### 2.3. Logistic Regression Analysis and Decision-Tree Analysis for Improvement of LFI

The independent factors associated with an improvement in LFI are summarized in Figure 2A. Group, sex, and ALBI grade were selected by stepwise regression, and the in-hospital exercise group and females were identified as independent factors associated with an improvement of LFI (Figure 2A).

To clarify the profile associated with an improvement in LFI, a decision-tree algorithm was created using three divergence variables and patients were classified into four profiles (Figure 2B). The group was identified as the initial divergence variable for changes in LFI. An improvement in LFI was seen in 74% of patients in the exercise group (Profile 1 in Figure 2B). On the other hand, an improvement in LFI was seen in 54% of patients in the non-exercise group. Among patients in the non-exercise group, sex was selected as the second divergence variable for changes in LFI. An improvement in LFI was seen in 73% of females and 41% of males (Profile 2 in Figure 2B). Among females, BCAA supplementation was selected as the third divergence variable for changes in LFI. An improvement in LFI was seen in 89% of patients with BCAA supplementation (Profile 3 in Figure 2B), while improvement in LFI was seen in 65% of patients with no BCAA supplementation (Profile 4 in Figure 2B).

In this analysis, the following factors were not identified as divergence variables for improvement of LFI: age, BMI, ALBI grade, TNM stage, presence of sarcopenia, VFA, and hospitalization period.

### 2.4. Differences in Patient Characteristics between the Exercise and Non-Exercise Groups after Propensity Score Matching

Patient characteristics after propensity score matching are summarized in Table 2. Propensity scores for all patients were estimated using the following baseline characteristics as covariates: age, sex, and BMI. No significant difference was seen in the etiology of liver disease, treatment of HCC, or TNM stage between the two groups. There was no significant difference in ALBI grade or the prevalence of BCAA supplementation between the two groups. No significant difference was seen in biochemical examinations between the exercise and the non-exercise groups, except for blood ammonia levels (Table 2).

There was no significant difference in LFI score and the prevalence of pre-frail/frail between the two groups. There was no significant difference in the presence of sarcopenia between the two groups. The hospitalization period was significantly shorter in the exercise group than in the non-exercise group. In the exercise group, the median exercise time was four days and the median exercise implementation rate was 83.0% (Table 2).

### 2.5. Differences in LFI Changes during Hospitalization between the Exercise and Non-Exercise Groups after Propensity Score Matching

The ΔLFI was significantly lower in the exercise group than in the non-exercise group, even after propensity score matching (Figure 3).

### 2.6. Logistic Regression Analysis and Decision-Tree Analysis for Improvement of LFI after Propensity Score Matching

Independent factors for improvement of LFI after propensity score matching are summarized in Figure 4A. Sex and group were selected by stepwise regression, and in-hospital exercise, and females were identified as independent factors associated with an improvement in LFI (Figure 4A).

To clarify the profile associated with an improvement in LFI, a decision-tree algorithm was created using four divergence variables and the patients were classified into five groups after propensity score matching (Figure 4B). Sex was identified as the initial divergence variable for changes in LFI. Among males, the group was the second divergence variable for changes in LFI. An improvement in LFI was seen in 69% of the exercise group (Profile 2 in Figure 4B). On the other hand, an improvement in LFI was seen in 42% of the non-exercise group (Profile 3 in Figure 4B). Among females, BCAA supplementation and the group were the second and third divergence variables for changes in LFI, respectively. In female patients with no BCAA supplementation, an improvement in LFI was seen in 93% of the exercise group (Profile 4 in Figure 4B). On the other hand, an improvement in LFI was seen in 65% of the non-exercise group (Profile 5 in Figure 4B).

In this analysis, the following factors were not identified as divergence variables for improvement of LFI: age, BMI, ALBI grade, TNM stage, presence of sarcopenia, VFA, and hospitalization period.

## 3. Discussion

This study demonstrated that in-hospital exercise significantly improved frailty in patients with HCC. Multivariate and decision-tree analyses also revealed that exercise was an independent factor for an improvement, which was confirmed by the propensity score matching analysis. In addition, BCAA supplementation was identified as a factor associated with the improvement of frailty in female patients with HCC. Thus, our study suggests that in-hospital exercise was effective in improving physical function in patients with HCC. In addition, BCAA supplementation may be effective for female patients with HCC.

In-hospital exercise improved frailty and was an important factor for the improvement of LFI. Our exercise program was a mixture of aerobic and resistance exercises. Previous research demonstrated that a mixed exercise program significantly improved handgrip strength in advanced cancer patients [16]. A mixed exercise program was also shown to improve the time of the chair stand test in elder patients during cancer treatments [17]. Furthermore, a mixed exercise program significantly improved balance function in patients with metastasized colorectal cancer [18]. In our study, we performed the exercises under the supervision of physical therapists. The supervised exercise led to a greater improvement in physical function than an unsupervised exercise in a meta-analysis of 34 randomized controlled trials (*n* = 4519) [19]. Thus, our results were in agreement with these previous reports and suggest that physical function in patients with HCC can be improved by in-hospital mixed exercise under supervision by physical therapists. In addition, a systematic review demonstrated that exercise improves physical frailty in patients with chronic liver disease [12]. Moreover, a meta-analysis showed that exercise is effective in improving frailty status in frail older individuals [20]. Thus, the beneficial effect of exercise on frailty seems to be not peculiar to patients with HCC, and in-hospital exercise may be recommended for any patients with frailty or risk of frailty.

In this study, sex was also identified as an independent factor for changes in LFI. Although LFI was improved in females without exercise, females have been reported to be at higher risk of developing incident sarcopenia than males [21]. In fact, being female was identified as an independent factor of muscle atrophy in patients with HCC in a previous report [4]. Although the reason why frailty was more improved in females than males in the non-exercise group remains unclear, adiponectin may be a possible mechanism. Adiponectin has been reported to have important metabolic functions, including insulin-sensitizing and anti-inflammatory effects [22]. It has been reported that testosterone negatively regulates the secretion of adiponectin from adipocytes, and circulating adiponectin levels show a sex disparity, being higher in females than in males [23]. Higher serum adiponectin levels are also known to be associated with better performance on frailty measures [24]. Thus, the difference in adiponectin levels may be one of the possible mechanisms of a sex discrepancy in the improvement of LFI.

In this study, BCAA supplementation was also associated with changes in LFI. The effects of BCAA supplementation on the improvement of LFI were evident in female patients and were confirmed in the propensity score matching analysis. It has been reported that there is a positive relationship between plasma BCAA concentration and muscle function [25]. Uojima et al. [26] demonstrated that BCAA supplementation improved low muscle strength in patients with chronic liver disease. Leucine is a BCAA that is known to have anabolic effects on cell signaling and protein synthesis in muscle [27], and it has been proposed that it augments resistance training-induced changes in body composition and performance [28]. In addition, BCAA supplementation improves muscle strength by promoting ammonia clearance from the blood in patients with chronic liver disease [26]. It remains unclear why the beneficial effect of BCAA supplementation was only seen in females. However, Leahy et al. [29] reported that BCAA supplementation reduced muscle soreness in females, but not in males, suggesting that this gender difference is related to dose-per-body weight differences between male and female subjects.

This study has several limitations. First, this was a retrospective observational study. Second, data were obtained from hospitals specialized in the treatment of HCC. Third, it remains unclear whether higher LFI is associated with poor prognosis in patients with HCC. Fourth, it is unclear if the level of LFI in patients with HCC is peculiar and we did not evaluate the effects of in-hospital exercise on the recurrence of HCC and post-treatment complications in this study. Thus, it is required to perform a multicenter prospective cohort study to investigate the impact of exercise on the frailty, the recurrence of HCC, post-treatment complications, and prognosis of patients with HCC.

In conclusion, we showed that in-hospital exercise improved frailty in patients with HCC, which was confirmed by the propensity score matching analysis. In addition, BCAA supplementation was a factor associated with the improvement of frailty in female patients with HCC. Thus, in-hospital exercise may improve physical function in patients with HCC. Moreover, BCAA supplementation may be beneficial for female patients with HCC.

## 4. Materials and Methods

### 4.1. Study Design

We performed a multicenter retrospective observational study to evaluate the effects of in-hospital exercise on LFI of patients with HCC. All authors had access to the study data and reviewed and approved the final manuscript.

### 4.2. Subjects

We enrolled 181 consecutive patients with HCC who are admitted for HCC treatment from December 2018 to March 2020. Hospitalized patients who met the following criteria were included in this study: (1) 20 years of age or older, (2) agreed with the evaluation of LFI as at admission, (3) were treated with hepatic arterial infusion chemotherapy, TACE, or small-molecule tyrosine kinase inhibitors, and (4) hospitalized for 6 days or more. Patients were excluded if they had: (1) hepatic encephalopathy grade 2–4 according to the West Haven Criteria [30], (2) risk of esophageal variceal rupture, (3) heart failure, or (4) respiratory failure.

Exercise was recommended to all patients by medical doctors, nurses, and physical therapists. Patients who agreed to exercise were classified as the exercise group (*n* = 114), and those who did not agree to exercise were classified as the non-exercise group (*n* = 67).

### 4.3. Exercise Regimen

To maintain physical function during hospitalization for HCC treatment, patients in the exercise group were treated with exercise individually under the supervision of a physical therapist certified for the rehabilitation of cancer patients. Six physical therapists delivered the exercise program. They were certified for the rehabilitation of cancer patients and had an average of 14.5 years (range 9–25 years) of clinical experience with hepatocellular carcinoma. Two (30%) of these physical therapists had Doctor of Philosophy (Ph.D.) degree qualifications. The oral agreements between the patient and the therapist were reinforced in writing in the form of a “treatment contract”. The exercise was initiated on the day after HCC treatment unless the patient had a fever of 38.0 °C or greater or liver failure. The frequency of exercise was 5 times/week. The exercise was performed in a rehabilitation room or hospital ward. The physical therapist attended the exercise covering the delivery of the exercise program. During the exercise, the therapist gave comments for the patients’ effort. The adherence to exercise was defined as the percentage of supervised exercise sessions completed. Besides this exercise program, there were no other exercise and functional tasks. Chemotherapy for HCC and nutritional therapy including BCAA supplementation were non-exercise components. The exercise consisted of the following four types of training according to the guidelines of the American College of Sports Medicine [31]. The exercise program was described using the Consensus on Exercise Reporting Template (CERT) by Slade et al. [32] (Appendix A).

#### 4.3.1. Stretching

Patients performed a series of stretches that targeted the muscles of the quadriceps femoris muscles, hamstrings, hip adductor muscles, gastrocnemius, back muscles, and shoulder muscles. Each stretch was held for 10–20 s. Patients progressed each stretch until a feeling of tightness and slight discomfort was felt [14,15]. The physical therapist showed a role model of the stretch and checked the stretching form, tightness, and discomfort. The total stretching time was 3–5 min (Appendix A).

#### 4.3.2. Resistance Training

The training included the following four generic exercises: (1) hip hinge movement (good-morning exercise), (2) towel air pull-down, (3) squats, and (4) calf raises. One set comprised of 10 repetitions, and a maximum of 3 sets was performed. Physical therapists adjusted the method of exercises and environment to be able to complete 10 repetitions for the exercise. Exercise load was own weight or own manual resistance. The total resistance training time was 5 min (Appendix A).

#### 4.3.3. Balance Training

Balance training was performed in both a one-leg stance or tandem stance using parallel bars and handrails. Patient keep position up to 1 min in one-legged stance and tandem. The tandem stance method maintains a posture while standing in a straight line. A one-leg stance maintaining a horizontal position in a one-legged position. These were done once on each side. The total balance training time was 5 min (Appendix A).

#### 4.3.4. Aerobic Training

Aerobic training was performed using a bicycle ergometer (Konami Sports Co., Ltd., Tokyo, Japan, or Senoh Corporation, Chiba, Japan), recumbent cross trainer (NuStep^®^, Senoh Corporation) [33], or by walking. The intensity of exercise was adjusted to maintain a subjective rating of perceived exertion of 11–13 points on the Borg scale [28]. The exercise load was increased according to the patient’s physical condition. The individual exercise intensity was adapted during each session by adjusting the load or the cycling speed so that the exercise goals were achieved. The exercise duration was gradually increased up to 15 min (Appendix A).

### 4.4. Nutrition and Diet Therapy

Nutritional care for patients was conducted according to the guidelines on nutritional management for Japanese patients with liver cirrhosis [34]. Details of these guidelines include: (1) energy requirements of 25–35 kcal/kg (ideal body weight) per day, (2) required protein intake of 1.0–1.5 g/kg/day (if there is no protein intolerance), (3) a required fat intake with a lipid energy ratio of 20–25%, and (4) a 200 kcal late evening snack as a divided meal as recommended by the Evidence-based Clinical Practice Guidelines for Liver Cirrhosis [35].

### 4.5. Branched-Chain Amino Acid Supplementation

Branched-chain amino acid (BCAA) supplementation (BCAA granules and BCAA-enriched nutrients) is an approved medication for decompensated liver cirrhosis in Japan. Thus, BCAA supplementation was administered to cirrhotic patients with hepatic encephalopathy or hypoalbuminemia, following the indication criteria [36].

### 4.6. Measurement of Liver Frailty Index

All patients underwent objective measurements of frailty as previously described [37]: (1) grip strength was evaluated in the subject’s dominant hand by the average of three measurements using a digital grip strength meter (DKK5401, Takei Machine Industry, Niigata, Japan), (2) chair stands with the subject’s arms folded across the chest were evaluated by the time it took for the patient to perform it five times, and (3) balance testing was evaluated by the time that the subject could balance in three positions (feet placed side-to-side, semi-tandem, and tandem). The maximum time for each test was 10 s. These three tests were administered by trained physical therapists. The LFI was calculated using the following equation [37] (calculator available at http://liverfrailtyindex.ucsf.edu): (−0.330 × gender-adjusted grip strength) + (−2.529 × number of chair stands per second) + (−0.040 × balance time) + 6 = LFI.

### 4.7. Changes in Liver Frailty Index

Changes in LFI (ΔLFI) were evaluated by the difference between the admission and the discharge days.

### 4.8. Measurement of Skeletal Muscle Index, and Visceral Fat Area

Skeletal muscle mass and visceral fat area (VFA) were evaluated using computed tomography (CT) scan images obtained before and after HCC treatment. CT images were taken for HCC assessment. The lower border of the third lumbar vertebra and the umbilical line were used as standard landmarks for measuring skeletal muscle mass and VFA, respectively. Skeletal muscle mass and VFA were measured using Image-J software version 1.80 (National Institutes of Health, Bethesda, MD, USA) [38] or Slice Omatic V 5.0 rev-8 software (Montreal, Quebec, Canada Tomovision) as previously described [39]. Skeletal muscle mass was normalized by the square of height, and the data were expressed as skeletal muscle index (SMI) [39].

### 4.9. Diagnosis of Sarcopenia

Sarcopenia was diagnosed according to the Japan Society of Hepatology (JSH) diagnostic criteria for sarcopenia in patients with liver disease [40]. According to the JSH criteria, patients with decreased grip strength were defined as those with grip strength <26 kg for males or <18 kg for females. Patients with decreased skeletal muscle mass were defined as those with SMI < 42 cm^2^/m^2^ for males or <38 cm^2^/m^2^ for females. Patients with decreased grip strength and decreased SMI were diagnosed with sarcopenia. The other patients were classified as having non-sarcopenia.

### 4.10. Diagnosis, Tumor Node Metastasis Staging, and Treatment of Hepatocellular Carcinoma

HCC was diagnosed using a tumor biopsy or a combination of tests for serum tumor markers, such as alpha-fetoprotein (AFP) and des-γ-carboxy prothrombin (DCP), and imaging procedures such as ultrasonography, CT, magnetic resonance imaging, and/or angiography. The clinical-stage of HCC was evaluated by TNM staging based on the Liver Cancer Study Group of Japan criteria [41]. Treatment for HCC was selected based on the evidence-based clinical practice guidelines for HCC from The Japan Society of Hepatology [42].

### 4.11. Biochemical Tests

Blood biochemical tests included serum levels of AFP, DCP, aspartate aminotransferase (AST), alanine aminotransferase (ALT), lactate dehydrogenase, alkaline phosphatase (ALP), gamma-glutamyl transpeptidase (GGT), cholinesterase, total protein, albumin, total bilirubin, blood urea nitrogen (BUN), creatinine, estimated glomerular filtration rate (eGFR), sodium, potassium, chlorine, total cholesterol, triglyceride, creatine kinase, glucose, hemoglobin A1c (HbA1c), ammonia, prothrombin activity, and prothrombin time international normalized ratio. In addition, red blood cell count, hemoglobin level, white blood cell count, platelet count, neutrophil count, lymphocytes, and neutrophil/lymphocyte ratio were measured.

### 4.12. Propensity Score Matching

Propensity scores for all patients were estimated with a logistic regression model using the following baseline characteristics as covariates: age, sex, and body mass index (BMI). A one-to-one nearest-neighbor matching algorithm with an optimal caliper of 0.2 without replacement was used to generate 118 pairs of patients, as previously described [43,44]. Thus, 118 patients with HCC (exercise group (*n* = 59) and non-exercise group (*n* = 59)) were analyzed.

### 4.13. Statistical Analysis

Data are expressed as median (interquartile range (IQR)), range, or number. All statistical analyses were performed using Statistical Analysis Software (JMP Pro version 15.0; SAS Institute, Cary, NC, USA). Differences between the exercise and non-exercise groups were analyzed using Wilcoxon rank-sum tests. The level of statistical significance was set at *p* < 0.05. Changes in LFI between admission and discharge were evaluated using the ΔLFI. In addition, independent factors associated with an improvement in LFI were analyzed using logistic regression analysis and decision-tree analysis, as previously described [15,45]. Furthermore, to overcome possible selection bias between the groups, we performed one-to-one propensity score matching as previously described [44]. We then performed similar analyses, such as Wilcoxon rank-sum tests, a logistic regression analysis, and a decision-tree analysis to investigate the effects of in-hospital exercise on frailty.

## 5. Conclusions

In-hospital exercise improved frailty in patients with HCC, which was confirmed by the propensity score matching analysis. In addition, BCAA supplementation was a factor associated with the improvement of frailty in female patients with HCC. Thus, in-hospital exercise may improve physical function in patients with HCC. Moreover, BCAA supplementation may be beneficial for female patients with HCC.

## Figures and Tables

**Figure 1 cancers-13-00194-f001:**
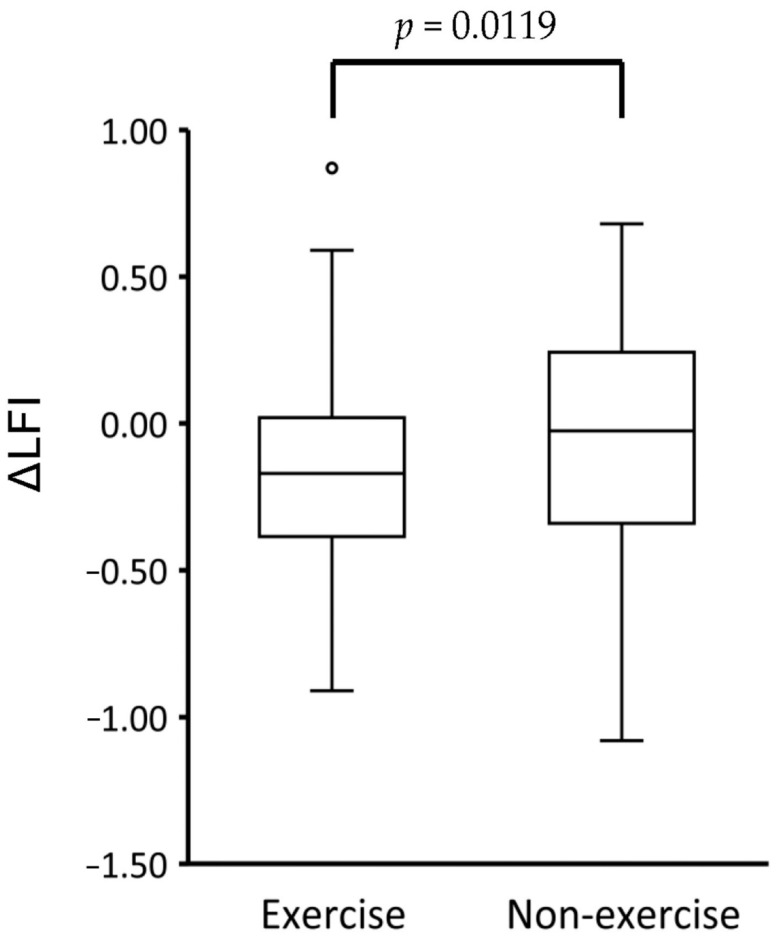
The difference in ΔLFI between the exercise group and the non-exercise group. Abbreviations: LFI; liver frailty index.

**Figure 2 cancers-13-00194-f002:**
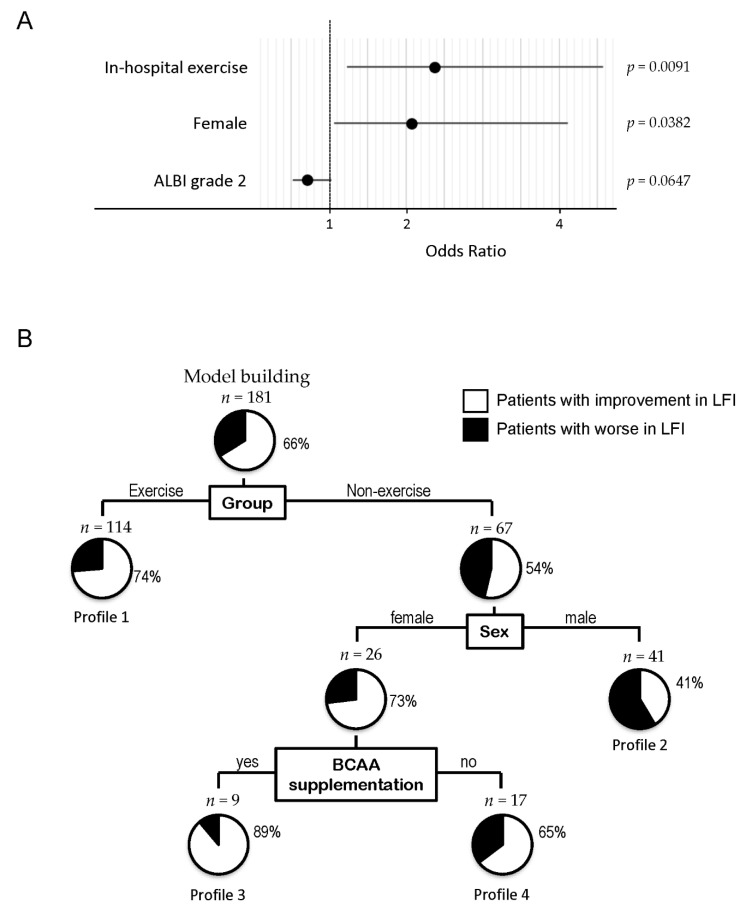
Factors associated with improvement of LFI. (**A**) Logistic regression analysis for improvement of LFI, (**B**) decision-tree algorithm for improvement of LFI in patients with HCC. The pie graphs indicate the proportion of patients with improvement in LFI (white) and patients without improvement in LFI (black). Abbreviations: LFI; liver frailty index, HCC; hepatocellular carcinoma.

**Figure 3 cancers-13-00194-f003:**
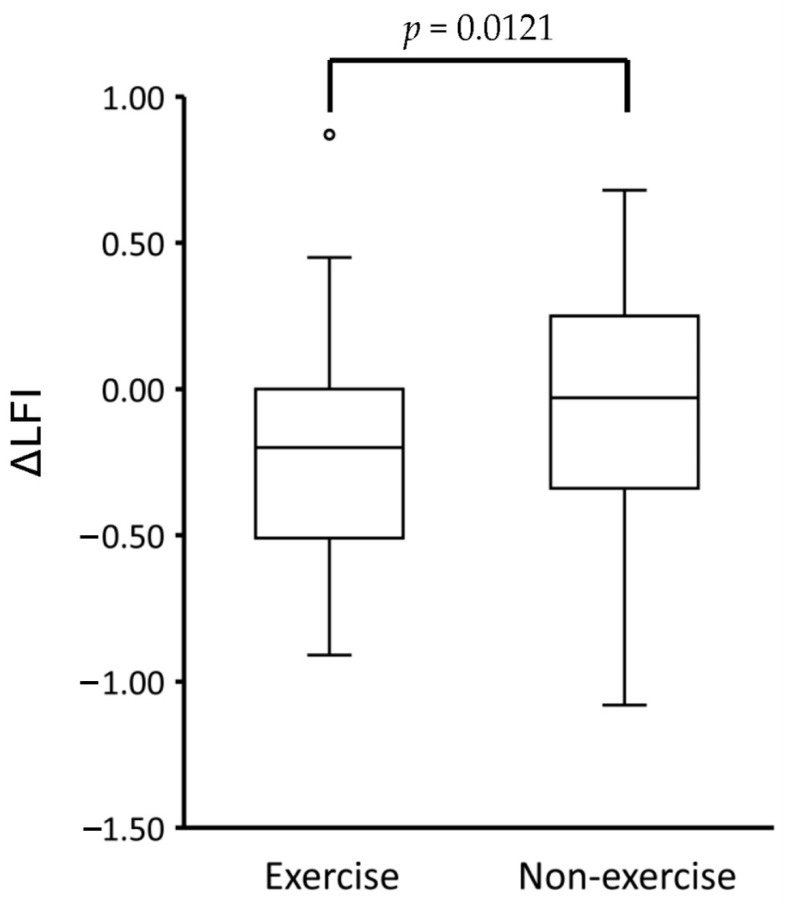
The difference of ΔLFI between the exercise group and the non-exercise group after propensity score matching. Abbreviations: LFI; liver frailty index.

**Figure 4 cancers-13-00194-f004:**
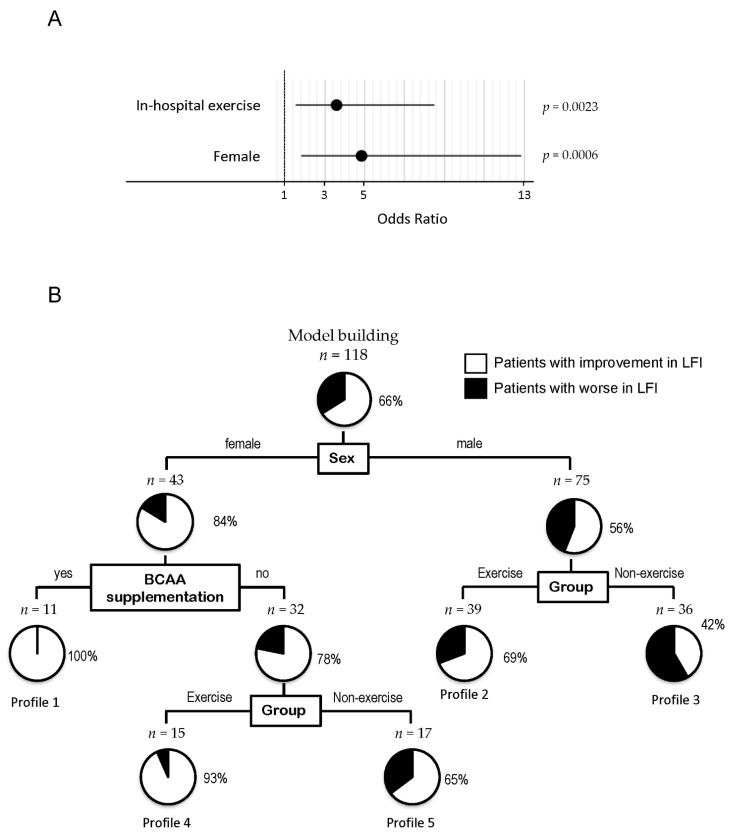
Factors associated with improvement of LFI after propensity score matching. (**A**) Logistic regression analysis for improvement of LFI after propensity score matching, (**B**) Decision-tree algorithm for improvement of LFI in patients with HCC after propensity score matching. The pie graphs indicate the proportion of patients with improvement in LFI (white) and patients without improvement in LFI (black). Abbreviations: LFI, liver frailty index; HCC, hepatocellular carcinoma.

**Table 1 cancers-13-00194-t001:** Patients’ characteristics.

Variable	Reference Value	Exercise		Non-Exercise		
		Median (IQR)	Range (Min–Max)	Median (IQR)	Range (Min–Max)	*p*-Value
Number	N/A	114	N/A	67	N/A	N/A
Age (years)	N/A	79 (74–83)	57–94	76 (70–81)	49–92	0.0200
Sex (female/male)	N/A	42.1%/57.9%(48/66)	N/A	38.8%/61.2%(26/41)	N/A	0.6629
Body mass index (kg/m^2^)	18.5–24.9	22.9 (21.0–25.2)	15.6–32.7	23.6 (21.5–25.8)	18.1–31.3	0.2118
Etiology of liver disease (Alcohol/HBV/HCV/NAFLD/Others)	N/A	7.9%/11.4%/57.0%/14.0%/9.7% (9/13/65/16/11)	N/A	16.4%/7.5%/58.2%/4.5%/13.4% (11/5/39/3/9)	N/A	0.1048
Treatment of HCC (HAIC/TACE/Sorafenib/Ramucirumab/Lenvatinib/Others)	N/A	3.5%/85.1%/1.8%/0.9%/7.9%/0.9% (4/97/2/1/9/1)	N/A	0.0%/98.5%/0.0%/0.0%/1.5%/0.0% (0/66/0/0/1/0)	N/A	0.1227
TNM stage (I/II/III/IVa/IVb)	N/A	16.7%/37.7%/35.1%/6.1%/4.4% (19/43/40/7/5)	N/A	13.4%/38.8%/31.3%/9.0%/7.5% (9/26/21/6/5)	N/A	0.7975
Albumin-bilirubin (ALBI) grade (1/2/3)	N/A	67.5%/28.1%/4.4% (77/32/5)	N/A	65.7%/32.8%/1.5% (44/22/1)	N/A	0.4943
BCAA supplementation (yes/no)	N/A	22.8%/77.2% (26/88)	N/A	35.8%/64.2% (24/43)	N/A	0.0587
LFI (score)	<3.20	3.84 (3.27–4.27)	2.37–5.48	3.57 (3.08–4.03)	1.95–5.51	0.0234
LFI (Robust/Pre-frail-Frail)	N/A	20.2%/79.8% (23/91)	N/A	28.4%/71.6% (19/48)	N/A	0.3368
Sarcopenia (Sarcopenia/Non-sarcopenia)	N/A	32.5%/67.5% (37/77)	N/A	22.4%/77.6% (15/52)	N/A	0.1484
Barthel Index (0–100)	N/A	100 (100–100)	50–100	100 (100–100)	55–100	0.0342
SMI (low/high)	N/A	49.1%/50.9% (56/58)	N/A	62.7%/37.3% (42/25)	N/A	0.0770
VFA (cm^2^)	N/A	86.9 (54.5–152.3)	9.7–310.9	100.4 (72.4–120.0)	17.6–206.5	0.6327
Hospitalization (days)	N/A	9 (9–11)	6–24	12 (11–13)	6–22	0.0001
Exercise days	N/A	4 (3–5)	1–13	N/A	N/A	N/A
Exercise implementation rate (%)	N/A	100.0 (75.0–100.0)	25.0–100.0	N/A	N/A	N/A
Biochemical examinations						
alpha-fetoprotein (AFP) (ng/mL)	≤10.0	10.0 (3.6–85.4)	1.0–64458.0	11.7 (3.3–275.3)	1.0–146556.0	0.8543
des-γ-carboxy prothrombin (DCP) (mAU/mL)	≤40.0	49.0 (22.0–239.0)	6.3–292689.0	110.5 (24.8–960.5)	1.1–138123.0	0.0600
AST (IU/L)	13–30	32 (25–43)	4–111	33 (26–47)	16–152	0.4198
ALT (IU/L)	10–30	23 (18–33)	5–126	23 (14–33)	6–139	0.4038
Lactate dehydrogenase (IU/L)	120–240	205 (183–234)	114–428	204 (189–229)	153–487	0.7743
ALP (IU/L)	115–359	317 (239–413)	150–3021	372 (252–495)	148–967	0.2476
GGT (IU/L)	13–64	47 (29–85)	12–659	53 (35–90)	10–421	0.5293
Cholinesterase (U/L)	201–421	208 (172–238)	73–411	214 (133–259)	63–380	0.9194
Total protein (g/dL)	6.6–8.1	7.1 (6.8–7.4)	5.9–8.7	7.2 (6.7–7.6)	5.9–8.2	0.3803
Albumin (g/dL)	4.1–5.1	3.8 (3.3–4.1)	2.0–4.7	3.8 (3.3–4.2)	2.6–4.6	0.7932
Total bilirubin (mg/dL)	0.20–1.20	0.70 (0.50–1.00)	0.30–2.90	0.80 (0.60–1.10)	0.30–3.20	0.2772
BUN (mg/dL)	8.0–22.0	18.2 (13.2–22.8)	7.0–59.0	17.0 (12.0–21.0)	10.0–53.0	0.2450
Creatinine (mg/dL)	0.65–1.07	0.82 (0.67–1.00)	0.44–5.11	0.79 (0.64–0.92)	0.41–3.80	0.3941
eGFR (mL/min/1.73 m^2^)	>90.0	63.5 (49.6–74.3)	7.0–140.9	66.5 (54.2–81.7)	9.5–116.3	0.0726
Sodium (mmol/L)	138–145	140 (139–142)	127–146	140 (139–142)	130–147	0.5804
Total cholesterol (mg/dL)	150–199	167 (144–193)	89–235	170 (145–198)	104–267	0.6482
Triglyceride (mg/dL)	35–149	93 (73–126)	29–305	97 (76–133)	27–301	0.2524
Creatine kinase (U/L)	59–248	88 (60–129)	17–656	81 (58–124)	31–338	0.5646
Glucose (mg/dL)	0–99	113 (97–135)	65–495	111 (99–152)	82–316	0.6741
HbA1c (%)	4.3–5.8	6.1 (5.7–6.6)	5.1–10.5	6.1 (5.7–6.8)	4.7–9.3	0.9942
Ammonia (µg/dL)	13–86	46 (35–67)	17–180	41 (31–70)	13–130	0.2780
Prothrombin activity (%)	80–120	93 (84–103)	30–130	96 (79–108)	11–130	0.8135
Hemoglobin (g/dL)	13.7–16.8	12.5 (10.8–13.6)	7.3–15.7	12.6 (10.5–13.9)	8.1–16.2	0.8994
White blood cell count (/µL)	3300–8600	4400 (3500–5450)	1000–8900	4600 (3300–5900)	500–8700	0.9104
Platelet count (× 10^3^/mm^3^)	15.8–34.8	134.0 (93.0–179.5)	62.0–368.0	129.0 (80.0–180.0)	40.0–419.0	0.4327
Neutrophil/Lymphocyte Ratio	0.86–2.77	2.46 (1.71–3.67)	0.62–11.17	2.44 (1.68–3.45)	0.50–11.71	0.5402

Note. Data are expressed as median (interquartile range (IQR)), range, or number. Abbreviations: N/A; not applicable, HBV; hepatitis B virus, HCV; hepatitis C virus, NAFLD; non-alcoholic fatty liver disease, HCC; hepatocellular carcinoma, HAIC; hepatic arterial infusion chemotherapy, TACE; transcatheter arterial chemoembolization, TNM; tumor nodes metastasis, BCAA; branched-chain amino acids, LFI; liver frailty index, SMI; skeletal muscle index, VFA; visceral fat area, AFP; alpha-fetoprotein, DCP; des-γ-carboxy prothrombin, AST; aspartate aminotransferase, ALT; alanine aminotransferase, ALP; alkaline phosphatase, GGT; gamma-glutamyl transpeptidase, BUN; blood urea nitrogen, eGFR; estimated glomerular filtration rate, HbA1c; hemoglobin A1c, PT–INR; prothrombin time international normalized ratio.

**Table 2 cancers-13-00194-t002:** Patients’ characteristics after propensity score matching.

Variable	Reference Value	Exercise		Non-Exercise		
		Median (IQR)	Range (Min–Max)	Median (IQR)	Range (Min–Max)	*p*-Value
Number	N/A	59	N/A	59	N/A	N/A
Age (years)	N/A	77 (73–81)	59–94	77 (72–81)	58–92	0.9227
Sex (female/male)	N/A	33.9%/66.1% (20/39)	N/A	39.0%/61.0% (23/36)	N/A	0.5661
Body mass index (kg/m^2^)	18.5–24.9	23.2 (21.0–25.9)	17.1–31.3	23.5 (21.4–25.1)	18.1–31.3	0.8866
Etiology of liver disease (Alcohol/HBV/HCV/NAFLD/Others)	N/A	6.8%/15.2%/50.9%/15.2%/11.9% (4/9/30/9/7)	N/A	18.6%/6.8%/55.9%/5.1%/13.6% (11/4//33/3/8)	N/A	0.0780
Treatment of HCC (HAIC/TACE/Sorafenib/Lenvatinib/Other)	N/A	1.7%/89.8%/1.7%/5.1%/1.7% (1/53/1/3/1)	N/A	0.0%/98.3%/0.0%/1.7%/0.0% (0/58/0/1/0)	N/A	0.3764
TNM stage (I/II/III/IVa/IVb)	N/A	15.2%/39.0%/33.9%/5.1%/6.8% (9/23/20/3/4)	N/A	10.1%/44.1%/33.9%/6.8%/5.1% (6/26/20/4/3)	N/A	0.8991
ALBI grade (1/2/3)	N/A	64.4%/30.5%/5.1% (38/18/3)	N/A	67.0%/28.8%/1.7% (41/17/1)	N/A	0.5648
BCAA supplementation (yes/no)	N/A	23.7%/76.3% (14/45)	N/A	32.2%/67.8% (19/40)	N/A	0.3051
LFI	<3.20	3.80 (3.22–4.30)	2.37–4.87	3.51 (3.08–4.10)	2.21–5.51	0.1230
Frailty determination by LFI (Robust/Pre-frail-Frail)	N/A	22.0%/78.0% (13/46)	N/A	28.8%/71.2% (17/42)	N/A	0.5382
Sarcopenia (Sarcopenia/Non-sarcopenia)	N/A	22.0%/78.0% (13/46)	N/A	25.4%/74.6% (15/44)	N/A	0.6652
Barthel Index (0–100)	N/A	100 (100–100)	65–100	100 (100–100)	55–100	0.2043
SMI (low/high)	N/A	49.2%/50.9% (29/30)	N/A	67.8%/32.2% (40/19)	N/A	0.0399
VFA (cm^2^)	N/A	96.1 (66.2–150.0)	10.1–252.0	98.3 (61.4–122.6)	17.6–206.5	0.5379
Hospitalization (days)	N/A	9 (9–11)	6–24	12 (11–13)	6–20	0.0001
Exercise days	N/A	4 (3–5)	1–13	N/A	N/A	N/A
Exercise implementation rate (%)	N/A	83.0 (75.0–100.0)	25.0–100.0	N/A	N/A	N/A
Biochemical examinations						
AFP (ng/mL)	≤10.0	11.3 (4.1–147.0)	1.0–64458.0	12.7 (2.8–267.0)	1.0–146556.0	0.5611
DCP (mAU/mL)	≤40.0	54.0 (22.0–450.0)	6.3–292689.0	107.0 (25.0–841.0)	1.1–86289.0	0.2876
AST (IU/L)	13–30	35 (26–44)	14–103	34 (26–47)	16–152	0.7284
ALT (IU/L)	10–30	28 (18–36)	5–126	23 (14–33)	8–139	0.1171
Lactate dehydrogenase (IU/L)	120–240	200 (182–243)	143–428	205 (189–229)	153–487	0.8230
ALP (IU/L)	115–359	316 (229–424)	150–3021	363 (239–447)	148–967	0.4625
GGT (IU/L)	13–64	52 (31–112)	13–659	53 (35–91)	10–421	0.6926
Cholinesterase (U/L)	201–421	218 (147–239)	82–411	214 (145–261)	63–380	0.9036
Total protein (g/dL)	6.6–8.1	7.1 (6.8–7.4)	5.9–8.7	7.2 (6.8–7.6)	5.9–8.2	0.2458
Albumin (g/dL)	4.1–5.1	3.8 (3.3–4.1)	2.0–4.4	3.9 (3.3–4.2)	2.6–4.6	0.5206
Total bilirubin (mg/dL)	0.20–1.20	0.80 (0.50–1.00)	0.30–2.90	0.80 (0.60–1.00)	0.30–3.20	0.4604
BUN (mg/dL)	8.0–22.0	18.5 (13.0–22.9)	7.0–59.0	17.0 (12.0–22.0)	10.0–53.0	0.2826
Creatinine (mg/dL)	0.65–1.07	0.82 (0.68–0.97)	0.44–2.32	0.80 (0.64–0.95)	0.41–3.80	0.7103
eGFR (mL/min/1.73 m^2^)	>90.0	65.8 (52.2–77.0)	21.9–140.9	64.6 (53.2–81.7)	9.5–116.3	0.6110
Sodium (mmol/L)	138–145	140 (139–142)	134–144	140 (139–142)	130–147	0.9718
Total cholesterol (mg/dL)	150–199	170 (145–196)	89–235	169 (145–206)	124–267	0.9024
Triglyceride (mg/dL)	35–149	93 (66–125)	29–305	99 (84–139)	27–301	0.1285
Creatine kinase (U/L)	59–248	97 (66–147)	17–264	82 (64–127)	31–338	0.4887
Glucose (mg/dL)	0–99	110 (96–134)	71–205	111 (99–149)	82–304	0.3351
HbA1c (%)	4.3–5.8	6.1 (5.7–6.6)	5.1–10.5	6.1 (5.7–6.8)	4.7–9.3	0.7513
Ammonia (µg/dL)	13–86	52 (39–85)	24–180	38 (30–65)	13–130	0.0105
Prothrombin activity (%)	80–120	92 (84–105)	30–130	97 (80–109)	11–130	0.7333
Hemoglobin (g/dL)	13.7–16.8	12.5 (10.8–13.3)	7.3–15.6	12.6 (10.6–13.5)	8.1–15.2	0.7103
White blood cell count (/µL)	3300–8600	4200 (3500–5300)	1000–8900	4700 (3300–5900)	500–8700	0.4805
Platelet count (× 103/mm^3^)	15.8–34.8	132.0 (94.0–174.0)	62.0–368.0	129.0 (90.0–178.0)	46.0–365.0	0.9828
Neutrophil/lymphocyte Ratio	0.86–2.77	2.37 (1.68–3.36)	0.62–11.2	2.44 (1.68–3.40)	0.50–11.71	0.8568

Note. Data are expressed as median (interquartile range (IQR)), range, or number. Abbreviations: N/A; not applicable, HBV; hepatitis B virus, HCV; hepatitis C virus, NAFLD; non-alcoholic fatty liver disease, HCC; hepatocellular carcinoma, HAIC; hepatic arterial infusion chemotherapy, TACE; transcatheter arterial chemoembolization, TNM; tumor nodes metastasis, BCAA; branched-chain amino acids, LFI; liver frailty index, SMI; skeletal muscle index, VFA; visceral fat area, AFP; alpha-fetoprotein, DCP; des-γ-carboxy prothrombin, AST; aspartate aminotransferase, ALT; alanine aminotransferase, ALP; alkaline phosphatase, GGT; gamma-glutamyl transpeptidase, BUN; blood urea nitrogen, eGFR; estimated glomerular filtration rate, HbA1c; hemoglobin A1c, PT–INR; prothrombin time international normalized ratio.

## Data Availability

The data presented in this study are available on request from the corresponding author. The data are not publicly available due to ethical issue.

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
