# Peer review of "Effects of In-Hospital Exercise on Frailty in Patients with Hepatocellular Carcinoma"

_cancers, 2021, doi:10.3390/cancers13020194_

Round 1
Reviewer 1 Report
The study appears to be quite pertinent. The Japanese in-hospital exercise for patients with cancer included in the health care service by the Japanese Ministry of Health, Labor, and Welfare will have a great impact on this population. However, there is no description of the exercise program itself, except that it consists of stretching, strength training, balance practice, and aerobic training [12]. Surprisingly, the paper does include a methods section, after the results section (?!), thus it is unclear if there is a previously published protocol of the study. The sections of the papers must be re-organized. "Although exercise intensity was the same in all hospitals, instruments, and protocols were not exactly the same in all institutions" this statement is not clear, regarding the means of controlling the intensity and the instruments used. The exercise program should be described using the Consensus on Exercise Reporting Template (CERT) by Slade et al. (2016). It is somehow described in reference 12, however, it doesn´t include enough information regarding the program itself and the motivational strategies. "Therapeutic exercise in hospitalized patients with cancer is an approved health care service covered by the Ministry of Health, Labour, and Welfare of Japan health insurance. Therefore, a randomized controlled study, which intentionally set the control group (non-exercise group) was contrary to medical ethics" I must strongly support this statement. However, it should be outlined in the ethics section. The American College of Sports Medicine reference [13] is missing.
Reviewer 2 Report
This article seems to be nicely written, and be of interest and informative.
It can be acceptable in Cancers.
Please consider to quote the following reports regarding frailty and liver.
- Life (Basel). 2020 May 24;10(5):76.
- Diagnostics (Basel). 2020 Jun 25;10(6):433.
Reviewer 3 Report
Authors described that Patients with HCC who had pretreatment in-hospital exercise was affected for decreasing of LFI and under propensity score matching analysis, Thus, i pretreatment in-hospital exercise may be beneficial for improving physical function in patients with HCC.
Major
1) LFI should decrease with exercise, so isn't it a result peculiar to HCC patients?
If level of LFI in patients with HCC is considered to be peculiar, it may be need to evaluate recurrence and post-treatment complications.
2) Why authors should low LFI cases be collected and evaluated? Isn't the patient prognosis worse for patients with higher LFI?
Round 2
Reviewer 3 Report
Authors described that Patients with HCC who had pretreatment in-hospital exercise was affected for decreasing of LFI and under propensity score matching analysis, Thus, i pretreatment in-hospital exercise may be beneficial for improving physical function in patients with HCC.
I can not find any response to my previous Qx about below on your revise manuscript,
LFI should decrease with exercise, so isn't it a result peculiar to HCC patients? If level of LFI in patients with HCC is considered to be peculiar, it may be need to evaluate recurrence and post-treatment complications.
Just let me know your comments.
Author Response
To REVIEWER 3
Thank you very much for your letter regarding our manuscript (cancers-994668). We appreciate your comments, which have helped us to improve our manuscript. In line with your comments, please find below our response.
Comment 1: LFI should decrease with exercise, so isn't it a result peculiar to HCC patients? If level of LFI in patients with HCC is considered to be peculiar, it may be need to evaluate recurrence and post-treatment complications.
Answer: We apologize for the insufficient response to your comments. Exercise improves physical frailty in patients with chronic liver disease [1] as well as older individuals [2]. Thus, the beneficial effect of exercise on frailty seems to be not peculiar to patients with HCC, and in-hospital exercise may be recommended for any patients with frailty or risk of frailty.
As you pointed out, we agree that it is important to evaluate the effects of in-hospital exercise on the recurrence of HCC and post-treatment complications. However, these data are not available in our database and, therefore, this issue was described as a limitation and future research theme as following: In this study, it is unclear if the level of LFI in patients with HCC is peculiar and we did not evaluate the effects of in-hospital exercise on the recurrence of HCC and post-treatment complications. Thus, it is required to perform a multicenter prospective cohort study to investigate the impact of exercise on the frailty, the recurrence of HCC, post-treatment complications, and prognosis of patients with HCC. In the revised manuscript, we have added these descriptions (line 227-231 and line 259-263). Again, we appreciate your comments, which have helped us to improve our manuscript.
References
1 Williams, F.R.; Berzigotti, A.; Lord, J.M.; Lai, J.C., Armstrong, M.J. Review article: impact of exercise on physical frailty in patients with chronic liver disease. Aliment Pharmacol Ther. 2019; 50: 988-1000.
2 Liao, C.D.; Lee, P.H.; Hsiao, D.J.; Huang, S.W.; Tsauo, J.Y.; Chen, H.C., Liou, T.H. Effects of Protein Supplementation Combined with Exercise Intervention on Frailty Indices, Body Composition, and Physical Function in Frail Older Adults. Nutrients. 2018; 10: 10:1916.

Round 3
Reviewer 3 Report
Dear authors
Thank you for your polite response. This is a very important and impressive paper for a liver surgeon like me who handles HCC. Furthermore liver surgeons want to know how preoperative improvement in frailty status affected for the patient's prognosis.
Finally, I will ask the Editor to judgement of Accept or not.